# ABCAttack: A Gradient-Free Optimization Black-Box Attack for Fooling Deep Image Classifiers

**DOI:** 10.3390/e24030412

**Published:** 2022-03-15

**Authors:** Han Cao, Chengxiang Si, Qindong Sun, Yanxiao Liu, Shancang Li, Prosanta Gope

**Affiliations:** 1Key Laboratory of Network Computing and Security, Xi’an University of Technology, Xi’an 710048, China; caohan@stu.xaut.edu.cn (H.C.); liuyanxiao@xaut.edu.cn (Y.L.); 2National Computer Network Emergency Response Technical Team/Coordination Center of China (CNCERT/CC), Beijing 100029, China; sichengxiang@cert.org.cn; 3School of Cyber Science and Engineering, Xi’an Jiaotong University, Xi’an 710049, China; 4The Department of Computer Science and Creative Technology, University of the West of England, Bristol BS16 1QY, UK; Shancang.Li@ieee.org; 5Department of Computer Science, University of Sheffield, Sheffield S10 2TN, UK; Prosanta.nitdgp@gmail.com

**Keywords:** deep neural networks, adversarial examples, image classification, information security, black-box attack

## Abstract

The vulnerability of deep neural network (DNN)-based systems makes them susceptible to adversarial perturbation and may cause classification task failure. In this work, we propose an adversarial attack model using the Artificial Bee Colony (ABC) algorithm to generate adversarial samples without the need for a further gradient evaluation and training of the substitute model, which can further improve the chance of task failure caused by adversarial perturbation. In untargeted attacks, the proposed method obtained 100%, 98.6%, and 90.00% success rates on the MNIST, CIFAR-10 and ImageNet datasets, respectively. The experimental results show that the proposed ABCAttack can not only obtain a high attack success rate with fewer queries in the black-box setting, but also break some existing defenses to a large extent, and is not limited by model structure or size, which provides further research directions for deep learning evasion attacks and defenses.

## 1. Introduction

In recent years, deep learning has developed rapidly and successfully applied to a number of complex problems such as image classification [1] and traffic classification [2]. However many studies on adversarial examples [3,4] have proclaimed the vulnerability of DNN models, which brings great security risks to the deployment of DNN models in practical applications. Even in a DNN-based system with communication security defenses [5], attackers can break it by adding adversarial perturbations to the transmitted data in advance [6,7]. For example, when a DNN model is used for traffic sign recognition in an automatic driving system, attackers can maliciously modify the captured road sign image, causing the model to give false inference, which leads to the decision system making wrong judgments. It greatly increases the probability of safety accidents. Therefore, the security problems caused by adversarial samples in machine learning system cannot be ignored.

The adversarial sample refers to a synthetic sample formed by adding subtle perturbation to the original sample, and the visual difference between the adversarial sample and the original sample is imperceptible to humans. The concept of adversarial samples was first proposed by Szegedy et al. [4]. Adversarial samples have a significant research space in the field of computer vision and images [8]. The methods of generating adversarial samples can be divided into white-box and black-box. The white-box attacks require the attacker to have full knowledge of the target model, such as parameters, structure, types, objective functions, etc. The clean sample can be modified by calculating the true gradient of the victim model to generate the adversarial image. The black-box attacks mean that the attacker can only access the model input and feedback. In the real world, the assumption of white-box attacks is difficult to satisfy. Therefore, researchers have explored black-box attacks.

At present, the existing black-box attacks are to break the target model with adversarial samples carefully crafted on the substitute model [9], gradient evaluation [10] or swarm intelligence. However, generating adversarial examples on the substitute model to attack the target model actually builds a remote hosting system, but requires access to the target model to gather a dataset to train the substitute model. These attacks add extra overhead for gathering data and training a substitute model. After that, the adversarial samples generated by the white-box attacks on the substitute model are used to attack the victim model which is based on the transferability of adversarial examples. However, the success rate is relatively low, especially in targeted attacks, and evaluating the gradient using finite difference requires a higher number of queries and computational cost. Attacks adopting warm intelligence can perform in different DNN models because these do not rely on model structure and other detailed information, and the crafted adversarial samples are random, which can break some of the existing state-of-the-art defense strategies to a certain extent [11,12].

Motivated by the above, we propose a black-box adversarial attack using Artificial Bee Colony (ABC), called ABCAttack. It formalizes the adversarial sample generation problem into an optimization problem, and uses the ABC algorithm to solve it. This method needs no gradient information in the process of generating the adversarial samples, so it is gradient-free. The performance of the algorithm is evaluated on MNIST, CIFAR-10 and ImageNet datasets. The results show that ABCAttack can obtain a better attack success rate with fewer queries. In addition, we also found that classes with similar appearance characteristics are more likely to be crafted to each other. Due to its gradient-free nature, ABCAttack is robust to some defenses. We performed untargeted attacks on models which have different structures and sizes, and it can still achieve a high success rate with fewer queries. This is because the process of generating adversarial samples does not rely on gradient calculation or gradient evaluation. Therefore, it has a strong applicability. The main contributions are summarized as:Aiming at reducing the reliance of white-box attacks on the attacker’s knowledge and overcoming the shortcomings of black-box attacks that evaluate gradients and train a substitute model to generate adversarial samples, a new black-box adversarial sample generation method, ABCAttack, is proposed.ABCAttack transforms adversarial sample generation into an optimization problem by using an improved search strategy to continuously iterate to obtain the adversarial image that is adversarial samples if successfully attacking the targeted model. Adversarial samples can be generated only by accessing the input and softmax probability of the model, without other detailed information, so gradient evaluation and training a substitute model are avoided, which can effectively improve the generation efficiency of adversarial samples.We evaluated the proposed ABCAttack using existing datasets: MNIST, CIFAR-10 and ImageNet. The results demonstrate the effectiveness and efficiency under the black-box setting, that is, in both targeted and untargeted attacks, ABCAttack uses fewer queries and lower time consumption to generate adversarial samples, which destroys the trained DNN models and greatly reduces its credibility.We further highlight the effectiveness of ABCAttack in defenses, namely defence-Gan, Stochastic Activation Pruning (SAP), Local Intrinsic Dimensionality (LID), non-differentiable input transformations and others. Although these defenses claim to be robust, our attack can still greatly reduce the accuracy of defense models. ABCAttack is gradient-free, so it has wide applicability, which means that as long as we obtain the input and feedback, we can successfully craft adversarial examples without caring about the specific details of the model.

In Section 2, we describes the work related to adversarial samples, and Section 3 introduces the standard ABC algorithm. Section 4 formalizes the description of adversarial samples and discusses the specific details of our proposed algorithm. Section 5 presents the experimental results and uses multiple indicators to evaluate the performance of the algorithm. Section 6 discusses the effect of population size and the maximum amount of change allowed by adversarial perturbation on attack performance. Finally, we summarize the work of this paper and look forward to the future work in Section 7.

## 2. Related Work

The emergence of adversarial samples provides an alarm for the security of ML systems and promotes research on both attacks and defenses. The difference between white-box and black-box attacks is that in a white-box setting, the attacker has all the knowledge of the target model.

Szegedy et al. [4] proposed L-BFGS, which uses box-constrained optimization to generate adversarial examples, and pointed out that adversarial examples are transferable, that is, in two models that perform the same task, the adversarial samples that affect one model usually affect the other model, even if the two models have different architectures or are trained on non-duplicate training sets. Goodfellow et al. [13] described the rationale of adversarial samples, explaining that the high-dimensional linearity of deep neural networks is the reason for the existence of adversarial samples, and proposed the Fast Gradient Symbol Algorithm (FGSM). Kurakin et al. [7] proposed I-FGSM on the basis of FGSM, and used a cell phone camera to take an image of a printed adversarial sample as the input of the image classification neural network. As a result, most of the captured images were still misclassified, which confirms the existence of adversarial samples in the real world. Nguyen et al. [14] used evolutionary algorithms or gradient ascent algorithms on a successfully trained CNN model to find special images that humans cannot recognize but the DNN classifies with high confidence. Moosavi-Dezfooli et al. [15] proposed an untargeted attack, DeepFool, which can calculate a smaller perturbation than FGSM. JSMA [16], as a typical $L_{0}$-norm white-box attack, introduced Saliency Map [17], which seeks to minimize the number of pixels modified. C&W [18] is generally regarded as one of the most powerful white-box attacks, supporting $L_{0}$-norm, $L_{2}$-norm and $L_{\infty }$-norm attacks, which break the effectiveness of defensive distillation [19] for most attacks.

In the black-box setting, the attacker can only access the input and feedback. In view of the transfer nature of adversarial samples, Papernot et al. trained a local model to substitute for the target DNN, using synthetical inputs generated by an attacker and labeled by the target DNN [9]. Then, they introduced reservoir sampling to greatly improve the accuracy and reduce the computational cost of the substitute learning technique [20]. Chen et al. [10] only accessed the model input and feedback information to evaluate the gradient using the finite difference, but a lot of queries are required. Bhagoji et al. [21] also used the finite difference to evaluate the gradient for black-box attacks, and Principal Component Analysis (PCA) and random grouping were introduced to reduce queries. Narodytska et al. [22] randomly selected a single pixel or a small number of pixels to add perturbation, and then used the idea of greedy local search to craft a small group of pixels to improve the attack efficiency. Brendel et al. [23] proposed a decision-based black-box attack, starting with a large adversarial perturbation, and then sought to reduce the perturbation while maintaining the adversarial nature. Ilyas et al. [24] defined three realistic threat models: the query-limited setting, the partial information setting and the label-only setting, and developed new attacks under these restrictive threat models, which demonstrates that ML systems are still vulnerable to attacks even under limited queries and information.

In addition, Su et al. [25,26] merely modified a single pixel of the original image using differential evolution to attack. GenAttack [11], AdversarialPSO [27,28] used genetic and particle swarm optimization algorithms, respectively, to make adversarial samples. Both attacks greatly reduce the number of queries. In addition, Liu et al. [12] put forward a method of crafting adversarial samples based on the population evolutionary algorithm, which is resistant to defense distillation. It is found that the adversarial samples reproduce the features of the data learned by the DNN model to a certain extent.

Recently, Moosavi-Dezfooli et al. [29] and Mopuri et al. [30] proved the existence of universal adversarial perturbations, which are both network-agnostic and image-agnostic. With the studies on adversarial samples, defense algorithms have also emerged, which hope to improve the robustness of the model adversarial samples [31,32,33,34].

## 3. ABC

Artificial bee colony is a meta-heuristic swarm intelligence optimization algorithm proposed by Professor *Karaboga* in 2005 [35]. It is inspired by the principles of a bee colony’s foraging behavior through information sharing and communication. A bee colony collects nectar efficiently and collaboratively. In the process, Employed Bees (EBs) collect the nectar and return to the hive and share information with onlooker bees through dance. Onlooker Bees (OBs) choose the nectar according to the observed dance. Under certain conditions, the employed bee is transformed into a Scout Bee (SB), which randomly searches for new nectar near the hive. This enables the entire bee colony to complete the work of collecting honey in a collaborative manner efficiently. Inspired by this, the minimum search model for a bee colony to realize swarm intelligence includes four elements: food source, employed bees, onlooker bees and scout bees and, among them, OBs and SBs are called unemployed bees. Each food source is a candidate solution to the optimization problem being solved. If the number of food sources is SN, the numbers of EBs and OBs are both SN/2. Under special circumstances, the individuals in the colony will transform into SBs to search for a new food source. Suppose the dimension of the problem to be solved is *d*, and the position of the food source *i* in the *t* iteration is $X_{i}^{t}=\left[x_{i1}^{t},x_{i2}^{t},\ldots ,x_{id}^{t}\right]$. The main steps of the ABC algorithm are as follows:

**Step 1: Initialization.** The ABC algorithm uses Equation (Equation 1) to initialize EN food sources, and calculate the fitness for every food source.
(1)$$x_{ij}=x_{ij}^{\min}+\mathrm{rand}\left(0,1\right)\times \left(x_{ij}^{\max}-x_{ij}^{\min}\right)$$
where $x_{ij}$ is the component of food source *i* in the *j*-th dimension, i∈{0,1,2,…,EN−1}, j∈{0,1,2,…,d−1}. $x_{ij}^{\min}$ and $x_{ij}^{\max}$ are lower limit and upper limit of the *j*-th component, respectively. rand(0,1) represents a floating-point random number obtained randomly in the range of [0, 1].

**Step 2: Sending EBs.** Each EB exploits a new food source according to Equation (Equation 2),
(2)$$x_{ij}^{t+1}=x_{ij}^{t}+\varphi_{ij}\left(x_{ij}^{t}-x_{kj}^{t}\right)$$
where i,k∈{0,1,2,…,EN−1}, k≠i, $\varphi_{ij}$ is a floating-point random number in the range [−1, 1]. After that, the fitness value is calculated for the new food source, and each $x_{i}^{t+1}$ is compared with the corresponding $x_{i}^{t}$ to keep the better one. It is judged whether to update the feasible solution.

**Step 3: Sending OBs.** EBs share the information with OBs, and the OBs selectively mine new food sources according to probability.
(3)$$fitness_{i}^{t+1}=\left\{\begin{array}{cc} \frac{1}{1+g_{i}^{t+1}} & g_{i}^{t+1}\geq 0\\ 1+\left|g_{i}^{t+1}\right| & g_{i}^{t+1}<0 \end{array}\right.$$
(4)$$P_{i}^{t+1}=\frac{fitness_{i}^{t+1}}{\sum_{i=0}^{EN-1}fitness_{i}^{t+1}}$$

In Equation (Equation 4), $fitness_{i}^{t+1}$ is the fitness value of the food source in the generation t+1, $P_{i}^{t+1}$ is probability of the *i*-th solution and $g_{i}$ is an objective function. The food sources to be exploited are selected according to $P_{i}$, the search formula is the same as step 2. After that, the remaining procedures are the same as sending EBs.

**Step 4: Sending SBs.** Assuming the quality of the food source has not improved after reaching the limit evaluation, the corresponding EB becomes a SB, the current food source is abandoned and a new food source is generated according to Equation (Equation 1).

The ABC algorithm is to continuously mine and explore the food sources by bees with three different responsibilities, iteratively executing steps 2 to 4 to obtain the final solution.

## 4. Methodology

### 4.1. Adversarial Sample

The adversarial sample can be formally described as follows:(5)$$x_{adv}=x+\delta$$
where *x* is the original image, and it can be classified as true label *y* by *f*. δ is the perturbation, $x_{adv}$ is a carefully crafted adversarial sample.

The attacks can be divided into targeted and untargeted attacks according to whether the attacker specifies the model’s prediction label for the adversarial sample. Given the classifier f(x) and the original sample *x*, the goal of the untargeted attack is for $x_{adv}$ to be classified by f(x) into any other label except *y*, that is, $f\left(x_{adv}\right)\neq f\left(x\right)=y$. The targeted attack is to specify the prediction label $y^{*}$ of the $x_{adv}$ before the algorithm runs, i.e., $f\left(x_{adv}\right)=y^{*}$. In contrast, targeted attacks are more difficult. $x_{adv}$ is an adversarial sample made by the attacker on the model $f_{1}\left(x\right)$. Typically, it still has an attack effect for another model $f_{2}\left(x\right)$ that performs the same task as $f_{1}\left(x\right)$, which is called the transferability of adversarial samples.

On the basis of a successfully attack, in order to ensure the information invariance of the whole vision, that is, a human can correctly classify the adversarial samples, the attack algorithms will limit the perturbations added. Generally, the following formula is used to limit the perturbation:(6)$$L_{p}={∥x-x_{adv}∥}_{p}$$

When p=0, it is the $L_{0}$-norm of the matrix, indicating the number of non-zero elements in a vector. This constraint limits the dimension of the modified vector, but there is no limit on the modification range of each dimension when the adversarial sample is crafted.When p=2, it is the $L_{2}$-norm of the matrix, representing the Euclidean distance between the original image and the adversarial sample. This restriction allows the algorithm to modify all dimensions, but limits the magnitude.When p=∞, it is the $L_{\infty }$-norm of the matrix, which represents the maximum amount of modification between the original sample and adversarial sample, that is, the maximum absolute value of the added perturbation, which can modify all dimensions.

### 4.2. Problem Description

The essence of adversarial example generation is to find the adjacent images that match the target within the clean image neighborhood. In this study, ABC is used to continuously search for better candidate solutions in the search space by minimizing the fitness function until an adversarial sample is found or termination condition is reached.

We consider the generation of adversarial examples in a black-box setting. The attacker does not have any knowledge of the model and can only access input and softmax output. Assuming that the target classification model $f:R^{d}\to {\left[0,1\right]}^{k}$, where *d* is the input dimension and *k* is the output dimension and is also the number of classes, and *f* returns the probability of each label. Assuming that the original image $x=\{x_{1},x_{2},\cdots ,x_{d}\}$ can be correctly classified as true label *y* by the model *f*. The purpose of the attack is to find the perturbed vector $x_{adv}$, and the prediction label of the $x_{adv}$ is not *y* for untargeted attacks, or equal to the pre-defined target label $y^{*}$ for targeted attacks. In addition, the attacker should constrain the $L_{p}$ distance between clean the adversarial samples as much as possible to ensure that the visual difference between them is as imperceptible as possible. In this paper, we limit the $L_{\infty }$ distance.

For untargeted attacks:(7)$$\underset{c\in \{1,\cdots ,k\}}{\arg\max}f{\left(x_{adv}\right)}_{c}\neq y\;\mathrm{such}\;\mathrm{that}\;{∥x-x_{adv}∥}_{\infty }\leq \delta$$

For targeted attacks, set $y^{*}$ as the target label:(8)$$\underset{c\in \{1,\cdots ,k\}}{\arg\max}f{\left(x_{adv}\right)}_{c}=y^{*}\mathrm{such}\;\mathrm{that}\;{∥x-x_{adv}∥}_{\infty }\leq \delta$$

### 4.3. ABCAttack Algorithm

ABC is used in many fields to solve optimization problems. This paper introduces ABCAttack that relies on ABC to solve the optimization problems of Equations (Equation 7) and (Equation 8) to craft adversarial images. The process of ABC to solve the problem is mainly initializing, exploiting EBs, calculating selection probability and exploiting OBs and SBs to iterate continuously to find the adversarial image. Assuming that x and $x_{adv}$ are the clean image and the corresponding adversarial image, respectively, x is correctly classified as the real label *y*, and the target label pre-defined by the attacker is y*. As shown in Figure 1, ABCAttack takes the original image as the input. According to the input domain of the target model, the clean image is first mapped to the input space of the model. It is assumed that the input space range of the model is [−0.5, 0.5], therefore, we use $x^{\prime }=\left(x/255.0-0.5\right)$ to complete the normalization, and add random disturbance to $x^{\prime }$ to initialize the food source. After that, we iteratively optimize to obtain the adversarial sample. After the attack is successful, we then map it to the RGB space to observe the visual difference from the clean image. The algorithm will terminate the loop in advance when the adversarial sample is crafted successfully. The adversarial sample generation will fail if the maximum for queries is reached.

In order to further control the change range of the original image during the algorithm iteration process, we define the maximum change $\delta_{\max}$. ABCAttack crafts adversarial samples under the $L_{\infty }$-norm. The $L_{\infty }$-norm is defined as the maximum change in $x_{adv}$ relative to x, expressed as $\delta_{\max}=\max(|x_{1}-x_{adv}^{1}\left|,\right|x_{2}-x_{adv}^{2}\left|,\cdots ,\right|x_{d}-x_{adv}^{d}\left|\right)$.

We use the clip operation to limit changes in each candidate solution. If we set $\delta_{\max}=B$, the upper and lower limits of the original image are UP and LP, respectively, and the next candidate solution is $x^{t+1}=clipByTensor\left(x^{t+1},clip\left(x-B,LP,UP\right),clip\left(x+B,LP,UP\right)\right)$, where clip is the truncation function clip(vector,min,max), and its purpose is to assign min to elements smaller than min in the vector, and assign max to elements larger than max.

For each image, we use Algorithm 1 to generate its adversarial sample. In ABCAttack, we respectively define fitness functions for targeted and untargeted attacks. In untargeted attacks, the object function g(·) is the probability $p_{cand}^{y}$ that the original image is predicted as true label *y*. In targeted attacks, the object function g(·) is $1-p_{cand}^{y^{*}}$, where $p_{cand}^{y^{*}}$ is the probability of the pre-defined target label $y^{*}$ of the current candidate solution. In ABCAttack, the ranges of $p_{cand}^{y}$ and $1-p_{cand}^{y^{*}}$ are both (0, 1), and there is no case where g(·) is less than 0, so we choose the fitness function as:(9)$$\mathrm{calFitness}\left(x_{\mathrm{cand}}\right)=g\left(x_{\mathrm{cand}}\right)=\left\{\begin{array}{c} p_{\mathrm{cand}}^{y},\mathrm{untargeted}\\ 1-p_{\mathrm{cand}}^{y^{*}},\mathrm{targeted} \end{array}\right.$$
**Algorithm 1** Adversarial attack-based ABCAttack.**Input:** original image x, the true label *y* and its probability $prob_{ori}$, target label $y^{*}$, number of food source SN, Limit number *l*, max change $\delta_{\max}$, the initialization amplitude limit factor α, upper bounds UP and lower bounds LP.**Output:**BestSolution1: Poputions←[]2: BestSolution,bestFit←x, 100,0003: aMin,aMax←clip($x-\delta_{\max},LP,UP$), clip($x+\delta_{\max},LP,UP$)4: EN←SN/2    // Initializing5: **for**
i=0 to EN−1
**do**6:     Poputions[i]=$x+U\left(-\alpha \times \delta_{\max},\alpha \times \delta_{\max}\right)$7:     Poputions[i]=clipByTensor$\left(Poputions[i],aMin,aMax\right)$8:     Poputions[i].fitness = calFitness(Poputions[i])9:     updateBestSolution(Poputions[i], BestSolution)    // Optimization10: **while** stopOption() **do**11:     **for**
i=0 to EN−1
**do**12:         next = clipByTensor(searchStrategy(*i*, 1), aMin, aMax)13:         fitness = calFitness(next)14:         greedySelection()15:         updateBestSolution(next, BestSolution)          Poputions.p = selectProbabilities()16:     **for**
i=0 to EN−1
**do**17:         **if**
Poputions[i].p>rand(0,1)
**then**18:            next = clipByTensor(searchStrategy(*i*, 2), aMin, aMax)19:            fitness = calFitness(next)20:            greedySelection()21:            updateBestSolution(next, BestSolution)22:     sendScoutBees()23:     updateBestSolution(Poputions, BestSolution)      **return**
BestSolution

Therefore, the fitness value for targeted and untargeted attacks decreases as the cycle goes on, and the smaller the fitness, the better the current solution.

It can be seen from Algorithm 1 that ABCAttack consists of two main parts: initialization and optimization. The searchStrategy() is the implementation of search strategy. The calFitness() function calculates the fitness of the candidate solution by Equation (Equation 9). The updateBestSolution() updates the current optimal solution, and stopOption() is a function that judges whether the current cycle is stopped according to the termination conditions. The termination conditions include the successful generation of adversarial samples and queries reaching the maximum. The specific details of the algorithm are as follows.


**(1) Initialization**


In this paper, the initialization of the standard ABC algorithm (Equation (Equation 1)) is not used, but the candidate solutions are initialized by uniform distribution, as shown in Equation (Equation 10):(10)$$x_{i}^{0}=x+U\left(-a\times \delta_{\max},a\times \delta_{\max}\right)\left(i=1,2,\cdots ,EN\right)$$

Define the number of EBs and OBs EN, the maximum modification $\delta_{\max}$, that is, $L_{\infty }=\delta_{\max}$, and the initialization amplitude limiting factor α.


**(2) Optimization**


Iteratively optimizing solving adversarial examples using ABCAttack can be divided into the following stages: sending EBs, probability calculation, sending OBs and sending SBs. In the basic ABC algorithm, the search strategies of the EBs and the SBs are the same, both of which are Equation (Equation 2). In this paper, we improve the search strategy of the EBs and OBs to enhance the optimization ability and convergence speed.

Sending EBs stage: According to the ratio of the fitness of the current optimal solution to probability of the original image classified into the true label, the candidate solutions are divided into two parts, $S_{1}$ and $S_{2}$, and different search strategies are used for generating new candidate solutions. The specific division method is as follows:(11)$$N=\max\left(\left(EN-1\right)\times \frac{bestFit}{prob_{\mathrm{ori}}}-1,1\right)$$
(12)$$S_{1},S_{2}=S\left[:N\right],S\left[N:\right]$$

The search method of the candidate solution subset $S_{1}$ is the same as Equation (Equation 2), and the candidate solution subset $S_{2}$ is applied to Equation (Equation 13) to generate next generation candidate solutions, which use the current optimal solution to guide optimization:(13)$$x_{i}^{next}=S^{best}+\varphi_{i}\left(x_{i}^{t}-x_{k}^{t}\right),\left(i\in N,\cdots ,EN,k\neq i\right)$$

It can be seen that in the initial iteration of the algorithm, assuming that there is an untargeted attack, and $bestFit=prob_{\mathrm{ori}}$, the number of EBs that use the original search strategy in Equation (Equation 2) to exploit is EN−2, and the number of EBs that use the current optimal solution to guide the exploitation is 2. As the algorithm runs, bestFit gradually decreases with the increase in iterations, the number of EBs using the original search strategy in Equation (Equation 2) to exploit is gradually reduced and using optimal guided searches will increase.

After generating the next generation candidate solution, the fitness is calculated and the final next generation candidate solution based on greedy selection in Equation (Equation 14) is determined. If the candidate solution is replaced, its trial is set to 1, otherwise it is trial plus 1. This is the greedySelection() in Algorithm 1. Equation (Equation 15) is applied to update the current optimal solution if the $fit_{i}^{t+1}$ is less than bestFit.
(14)$$x_{i}^{t+1},{trial}_{i}^{t+1}=\left\{\begin{array}{c} x_{i}^{t},\;\mathrm{trial}{\;}_{i}^{t}+1,\;\mathrm{if}\;fit_{i}^{next}>fit_{i}^{t}\\ x_{i}^{\mathrm{next}\;},1,\;\mathrm{if}\;fit_{i}^{\mathrm{next}\;}\leq fit_{i}^{t} \end{array}\right.$$
(15)$$bestFit=\left\{\begin{array}{c} bestFit,\;\mathrm{if}\;{fit}_{i}^{t+1}>bestFit\\ {fit}_{i}^{t+1},\;\mathrm{if}\;fit_{i}^{t+1}\leq bestFit \end{array}\right.$$

Probability calculation and sending OBs stage: Calculate the selection probability for each candidate solution according to Equation (Equation 4), that is, the selectProbabilities() function in Algorithm 1. Select the candidate solution that satisfies the rand(0,1)<Popution[i].p condition for further exploitation. First, randomly select (rate×d)/2+1 elements with the probability rate for adaptive optimal guided local optimization. The rate can be obtained from Equation (Equation 16):(16)$$rate=\max\left(\frac{MaxEvaluation-evalCount}{MaxEvaluation},0\right)$$
where MaxEvaluation is the maximum queries, evalCount represents queries currently used. The rate shows a decreasing trend as the algorithm continues to iterate. Therefore, when the current optimal solution is closer to the attack target, the number of modified pixels should be reduced as much as possible to enhance the ability of local optimization. Let Points be the selected elements indexes set, and use the following equation to search new solutions for the sending OBs stage:(17)$$\begin{array}{cc} x_{i}^{next}\left[Points\right]= & S^{best}\left[Points\right]+\varphi_{i}\times \\ & \left(x_{i}^{t}\left[Points\right]-x_{k}^{t}\left[Points\right]\right),i\neq k. \end{array}$$

Then, calculating the corresponding fitness, carry out greedy selection according to Equation (Equation 14), and update the current optimal solution as seen in Equation (Equation 15).

Sending SBs stage: The numbers of evaluations of all current candidate solutions are checked. If the trial corresponding to a certain candidate solution is greater than the limitTrial, the current candidate solution is discarded, and a new candidate solution is generated by Equation (Equation 10).

The above steps are carried out iteratively until the adversarial sample is crafted or the termination condition of the algorithm is reached. Equation (Equation 2) generates a new food source near the randomly selected individuals, which has strong global search ability and can maintain the diversity of the population, but the convergence speed is slow. In the optimal solution guided search strategy, new food sources are generated near the optimal individual, which has strong local search ability, but it easily falls into local optimization. In the EB search stage, this paper adopts the combination of the original ABC search strategy (Equation (Equation 2)) and the search strategy guided by the current optimal solution (Equation (Equation 13)), which can ensure adjusting the population diversity and search ability according to the current optimal solution. We also compare the hybrid search strategy in this paper with the search strategy using only Equation (Equation 2) or Equation (Equation 13) on MNIST and CIFAR-10 datasets. With the same parameters, whether an untargeted or targeted attack, we prove that the algorithm can effectively improve the convergence speed, and the maximum reduction in average queries is more than 300, although this is limited. However, for lack of space, we will not show the specific experimental results in the experimental part.

## 5. Experimental Verification and Result Analysis

### 5.1. Experimental Dataset and Environment

We use MNIST [36], CIFAR-10 [37] and ImageNet [38] datasets to do comparative experiments to verify the performance of ABCAttack in ZOO [10], C&W [18], AdversitiaoPSO [27], GenAttack [11] and other methods. The models used by MNIST and CIFAR-10 are the same as those used in ZOO, C&W, AdversitiaoPSO and GenAttack. The specific details of the models can be found in [18], and their accuracies are 99.51% and 78.91% on the test datasets for MNIST and CIFAR-10, respectively. ImageNet uses the InceptionV3 model as the target attack model.

The experimental data used in this paper are the same as AdversitiaoPSO for MNIST and CIFAR-10. Under untargeted attacks, the first 1000 correctly predicted test samples in MNIST and CIFAR-10 are used to generate adversarial samples, while under targeted attacks, the first 111 test samples with correct prediction are used to attack. For one image, the pre-specified label of the targeted attack is set to all labels except the real label, and nine targeted attacks are performed. Therefore, the total number of targeted attacks is 111×9=999. The ImageNet dataset has 1000 classes. In this paper, 500 random samples with correct classifications are selected for attack. The following experiments on the MNIST and CIFAR-10 datasets were completed on a desktop computer with Intel Core i5-8400 @2.80 GHz CPU and 16 GB RAM, and for the experiments on the ImageNet, we completed the experiments on a machine with Intel Core i7-7700 @3.60 GHz × 8 CPU and GeForce CTX 1080Ti GPU.

In order to evaluate the performance of the algorithm, this paper introduces several metrics to measure the effectiveness of ABCAttack:

Attack success rate: ASR refers to the proportion of successful adversarial samples in all samples.

Average queries: We record an image individually input into the model as a query. The query count statistics are calculated only for successful attacks. The AvgQueries refers to the sum of the queries required for each successful adversarial example divided by the number of successful adversarial examples.

At the same time, we also calculated the $L_{2}$-norm of the adversarial perturbation, and also aimed to calculate the $AvgL_{2}$ on the successful sample. However, we pay more attention to the AvgQueries and the ASR.

### 5.2. Parameter Analysis

This attack first determines the number of food sources SN. We set the number of food sources to start from 10, with a step size of 2 to 30, perform untargeted attacks on the MNIST, CIFAR-10 and ImageNet datasets and observe the AvgQueries. We first describe the attack success rate, that when SN=10 and SN=12, respectively, the attack success rate is small, 92.9% and 99.5% on MNIST(α=1.0), 94.7% and 98.5% on CIFAR-10 (α=1.0), 76.67% and 84.67% on ImageNet. When SN≥14, the attack success rate of the algorithm tends to be stable. After that, the accuracy will fluctuate slightly as a result of some random factors in the algorithm. This is a normal phenomenon, because the algorithm contains randomness, but the change in accuracy is very slight. For example, the success rate on MNIST is between 99.9% and 100%. Figure 2 shows the relationship between SN and AvgQueries on MNIST, CIFAR-10 and ImageNet datasets.

It can be seen from the Figure 2 that, at the beginning, the AvgQueries decrease with the increase in SN. However, when it increases to a certain value, the AvgQueries increase slowly. When α=1.0, the optimal number of food sources is 30, 16 and 12 and the ASR is 99.9% 98.5% and 84.67% and, when α=0.5, they are 28, 24 and 16 and the ASR is 100%, 98.4% and 86.67%, respectively on MNIST, CIFAR-10 and ImageNet. However, with the increase in SN, the average $L_{2}$-norm also increases, even if it does not increase significantly (α=1.0, the average $L_{2}$-norms of MNIST and CIFAR-10 increase by 0.2526 and 0.0545. For α=0.5, the difference is 0.3605 and 0.1547, respectively). The trend of AvgQueries on ImageNet is not obvious, because when SN=22, the AvgQueries have a significant increase, and the other points are relatively stable after removing this point. Although the AvgQueries at this point have increased, the ASR has not changed much. Considering the ASR, AvgQueries and computing load, we finally choose SN as 20.

Equation (Equation 10) is used to initialize the candidate solution, where α is closely related to the $L_{2}$-norm of the initial candidate solution. According to the above experimental results of SN, we fixed SN at 20, and increased α from 0.1 to 1.0 in steps of 0.1. Figure 3 shows the results of untargeted attacks on MNIST, CIFAR-10 and ImageNet datasets, including the AvgQueries and $AvgL_{2}$ under different α. In MNIST and CIFAR-10 dataset experiments, we first show that while α changes, the ASR does not fluctuate much. When α=0.8 and α=0.9, the ASR of MNIST is 99.9%, and in other cases it is 100%. The ASR of CIFAR-10 is maintained at 98.3–98.5%. It can be seen from Figure 3 that the AvgQueries decrease monotonically with the increase in α, while the average $L_{2}$-norm increases monotonously. The following conclusions can be drawn: the $L_{2}$-norm of the initial candidate solution is directly proportional to the final adversarial perturbation $L_{2}$-norm, and is inversely proportional to AvgQueries used for successful attacks. The smaller the $L_{2}$-norm of the initial candidate solution is, the smaller the $L_{2}$-norm of the final adversarial perturbation that will be crafted, but the AvgQueries used for successful attack will also increase. On the contrary, the larger the $L_{2}$-norm of the initial candidate solution is, the AvgQueries used for successful attack are correspondingly lower.

However, the experimental results for the ImageNet dataset are not obvious. When α=0.3 and α=1.0, the AvgQueries decrease because the ASR of these two data points decreases, which is 83.33% and 84.67%, respectively. Compared with other points, the ASR of 89.33–92.00% is lower, resulting in low AvgQueries. However, unlike the MNIST and CIFAR-10, the AvgQueries and $AvgL_{2}$ have no particularly obvious trend on ImageNet. The $AvgL_{2}$ shows a downward trend from 0.1–0.5. However, the $AvgL_{2}$ shows a downward trend from 0.6–0.9, and the downward trend is not continuous. We think this is partly due to the increase in AvgQueries, because, in general, the more iterations, the more the $L_{2}$-norm will also increase. Considering ASR, AvgQueries and $AvgL_{2}$, we can see that α=0.5 is a better choice in the ImageNet dataset.

Parameter setting: In black-box attacks, the $L_{2}$-norm of perturbation cannot be balanced with the query cost. We can only try to trade off between them. Therefore, according to the above experimental results, we set SN=20, α=0.5, so the number of EBs and OBs is 10, and limitTrial=10. In MNIST and CIFAR-10, the MaxEvaluation is set to 10,000, and for ImageNet, the MaxEvaluation is set to 30,000. The image size of the MNIST is 28×28, the maximum change $\delta_{max}=0.3$, the image size is 32×32×3 in CIFAR-10 and 299×299×3 in ImageNet and the maximum change in these is $\delta_{max}=0.05$ When the model input range is [0, 1] or [−0.5, 0.5], and if the model input range is [0, 255], the $\delta_{max}=0.05\times 255=12$.

### 5.3. Comparison and Analysis of Experimental Results

#### 5.3.1. Attack on MNIST, CIFAR-10 Classification Models

As can be seen in Table 1, ABCAttack uses 629 and 1695 average queries in both untargeted and targeted attacks on the MNIST dataset, achieving 100% and 94.89% attack success rates, uses 330 and 851 average queries on the CIFAR-10 dataset, achieving 98.6% and 82.3% attack success rates, respectively. Compared with ZOO that uses 384,000 average queries, the average queries of untargeted attacks are reduced by about 609 times, and the average queries of targeted attacks are reduced by about 255 times in ABCAttack on the MNIST dataset. Compared with C&W that uses 4650 average queries, the average queries of ABCAttack are reduced by 6.3 times in untargeted attacks, and the average queries are reduced by about 1.74 times in targeted attacks. It is obvious from the table that this phenomenon is the same in the CIFAR-10 dataset. The average queries used by ZOO and C&W are several times that of ABCAttack, whether it is an untargeted or targeted attack.

In the experiment with MNIST, although the AvgQueries of ABCAttack are slightly lower than those of AdversarialPSO (ABCAttack: 629, AdversarialPSO: 593), the ASR of ABCAttack is better than that of AdversarialPSO. In targeted attacks, ABCAttack achieves a 94.89% attack success rate, and the average queries used are lower. In the experiment with CIFAR-10, only the ASR and $AvgL_{2}$ of untargeted attacks are slightly lower than the $AvgL_{2}$ (ABCAttack: 98.6%, 1.64319, AdversarialPSO: 99.6%, 1.4140). When α=0.1, in the MNIST experiment, the success rates of both SWISS and ABCAttack are 100%, and the $AvgL_{2}$ is comparable, but the average queries used by SWISS are about 2.47 times those of ABCAttack. However, in the targeted attack, the $AvgL_{2}$ of the adversarial samples generated by ABCAttack is 1 larger than that of SWISS. ABCAttack has more advantages in attack success rate and average queries. When α=0.1, in the CIFAR-10 dataset, only ABCAttack’s ASR (98.4%) is lower than that of SWISS (99.6%), and the rest are the best.

Secondly, in MNIST and CIFAR-10 experiments, Substitute+FGSM and Substitute+C&W both use white-box attacks to generate adversarial samples on the trained substitute models to attack the target model. It can be seen from Table 1 that the ASR needs to be improved. In the MNIST dataset, transferability reaches 84.24% for the first substitute DNN and 78.72% for the second, with input variations of $\delta_{max}=0.3$[9]. The ASR of transfer-based attacks depends on similarity between the model generating adversarial samples and the target model. With the same model structure, the more possibilities there are for successful transfer. However, the ASR with different structures is lower, especially in targeted attacks, because the decision boundaries of different models may be quite different, which cannot guarantee the success of target attacks. How to improve the transferability of adversarial samples is also a hot issue in current research.

Table 1 shows that only the $AvgL_{2}$ between the adversarial samples generated by ZOO and the original samples is the smallest, but the AvgQueries used are the largest. ABCAttack has more advantages in running time and queries.

#### 5.3.2. Attacks on Large-Scale Image Dataset ImageNet

The attack results of the large-scale ImageNet dataset are shown in Table 2. The target models of ZOO, AdversarialPSO and SWISS are all InceptionV3. Our attack algorithm also takes VGG-16 and VGG-19 as the target model to facilitate comparison with the UAP algorithm. The input of the InceptionV3 model is 299×299×3 or 224×224×3, and the input of VGG-16, VGG-19 and MobileNet is 224×224×3. In this experiment, we do not use dimension reduction or hierarchical attack to improve the attack efficiency. Therefore, the perturbation dimension is the same size as the image size. Parameters SN and α are still set to 20 and 0.5, and the maximum for queries is limited to 30,000. The first 500 randomly selected images predicted correctly by InceptionV3 in the ImageNet dataset are used as the original images to be input into the ABCAttack algorithm. The ASR and AvgQueries are 90.00% and 2759, respectively. In contrast, ZOO used 1,280,000 to produce adversarial samples with $AvgL_{2}$ of 1.19916 on the ImageNet dataset, and the ASR was as high as 88.9%, while the AdversarialPSO only used 2833 average queries, but the ASR was 82.00%. The $L_{2}$-norm in ABCAttack is much larger than that of ZOO and AdversarialPSO. However, the difference between the adversarial samples generated by this attack algorithm and the original samples cannot be perceived by human eyes, as shown in Figure 4. The main reason for the larger $L_{2}$-norm of the adversarial samples generated by ABCAttack is the larger perturbation dimension. ABCAttack used 2759 AvgQueries to achieve 90.00% ASR, while ZOO used 1,280,000 average queries, which is an order of magnitude difference. Compared with AdversarialPSO, our algorithm is about 70 fewer queries than that of AdversarialPSO, but has higher success rate, and the difference in AvgQueries is not significant, which is also related to the experimental samples.

Since different models have inconsistent processing schemes for input images, we set $\delta_{max}$ according to the model input. When the model input is between [0, 1], we set $\delta_{max}=0.05$, and when the model input is in the [−1, 1] range, we set $\delta_{max}$ to 0.1, but when the model input is [0, 255], we set it to 12, but this is basically consistent for the maximum change in the image in RGB space.

The UAP algorithm generates a universal perturbation without generating for each image. The accuracies of UAP [29] on VGG-16 and VGG-19 models are 78.3% and 77.8%. Some of the training images are used to craft universal perturbations, and [30] shows that universal perturbations can be crafted with non-training set images, which removes the limitation of knowing part of the training set data for universal perturbation generation, but as a cost, ASR decreases significantly. Tran-TAP and Tran-UAP [39] adopt transfer-based attacks. It can be seen that the ASR from VGG-16 to Inc-V3 is 76.1% and 68.8%. Although UAP is still transferable, its ASR is reduced. The ASR is only 34.3% when adversarial samples generated by Inc-V3-FGSM transfer to ResNet-V2. The difference in model structure is the bottleneck to improve the ASR of transfer-based attacks.

In this study, our ABCAttack (a=0.5) could obtain 99.6%, 99.0% and 100% success rates in VGG-16, VGG-19 and MobileNet, respectively, with average queries of 1839, 1501 and 568 when the maximum query MaxEvaluation is set to 30000. The attack success rate is significantly higher than that of the UAP algorithm.

In terms of attack cost time, ZOO needs to evaluate the gradient. For large-scale datasets, multi-dimensional gradient evaluation is quite time-consuming even with dimensionality reduction, so the attack time cost of the ZOO method is high. Compared with AdversarialPSO on the ImageNet dataset, we have a slight advantage in its average queries and average time spent, but we have an 8% increase in attack success rate, and regardless of whether we use VGG-16, VGG19, MobileNet or InceptionV3, our attack success rate is over 90.00%.

We conducted experiments with a greater query limit MaxEvaluation = 50,000. The experimental results are shown in Table 2. From the perspective of attack success rate, even if the maximum query MaxEvaluation is increased by 20,000, the increase in attack success rate is only slight. When VGG-16 and VGG-19 are used as target models, when the maximum query increases to 50,000, the attack success rate increases by 0.2% and 0.6%, respectively. When taking InceptionV3 as the target model, the attack success rate did not rise when the image size was 224. Even when the maximum query is increased to 70,000, a few samples still cannot obtain the corresponding adversarial samples. Compared with the maximum limit change $\delta_{\max}$, increasing the maximum limit query MaxEvaluation will not significantly improve the attack success rate, and will lead to an increase in the average queries. Similarly, we intentionally increased the MaxEvaluation when attacking the defense strategy, and the results can be seen in Table 3 and are consistent with the results on ImageNet.

However, we did not do any optimization in the ImageNet dataset, so the quality of generated adversarial images is not high. Improving the quality of adversarial samples, reducing the dimensions of algorithm input and reducing the average queries in a large-scale dataset in a black-box setting that does not rely on gradient calculations will be our future focus.

#### 5.3.3. Targeted Attack Analysis

This experiment selects the first 12 images of each class in the MNIST and CIFAR-10 datasets to carry out a targeted attack, and each image uses the remaining class as the pre-set label of the targeted attack. Therefore, 12×10×9=1080 attacks are executed. The ASR is 93.98% and 83.24%, and the corresponding AvgQueries and $AvgL_{2}$ are 1744 and 915, and 4.844 and 1.911, respectively. Figure 5 and Figure 6 show the analysis of the experimental results of a targeted attack on MNIST and CIFAR-10 datasets, respectively, where (a) is the heat map of the number of successful attacks, and the total number of attacks for each square is 12. Panel (b) shows the corresponding average queries. It can be seen from the figure that the original images predicted as auto and truck in the CIFAR-10 dataset are more difficult to be made into adversarial samples, while some samples are easier to be crafted adversarial images to attack models.

In the MNIST dataset, although the ASR is higher, when the pre-specified label is 0, the AvgQueries used for successful attack is more, which means that, in comparison, the clean samples of any other class are more difficult to be made into adversarial samples predicted to be 0. Compared with CIFAR-10, the AvgQueries used for successful attack in the MNIST dataset are more, and the heap map of both of them has slight symmetry, which is more obvious in CIFAR-10. In (c), the images on the main diagonal are the original images, and the other images are the generated corresponding adversarial images. It can be seen that there is a slight difference between the adversarial images generated by ABCAttack and the original images. However, the model classifies 10 different images into 10 different classes which can be correctly recognized as the same labels by a human. This further proves that DNN models still have many vulnerabilities and defects.

#### 5.3.4. Untargeted Attack Analysis

We select the first 100 correct prediction samples in each class of MNIST and CIFAR-10, and 10×100=1000 images are selected to craft untargeted adversarial images. The AvgQueries using 664 and 295 can achieve 99.9% and 98.3% ASR. Figure 7 and Figure 8 show the statistics of untargeted attacks on MNIST and CIFAR-10 datasets, respectively. Panel (a) is the relationship between the true labels of original images and predicted labels of corresponding adversarial images in the untargeted attack. Each node in (a) represents a class, and the node size indicates the number of adversarial samples that are predicted to be in the class. The larger the node, the more crafted adversarial images are predicted to be in the class, the color of the edge is consistent with the target node and the thickness of the edge represents the number of original–predicted label of adversarial image pairs. More detailed results can be seen in (b), which shows the relationship between the true class and predicted labels of the adversarial samples.

From Figure 7a and Figure 8a, it can be seen that on MNIST and CIFAR-10 datasets, the ABCAttack more easily creates adversarial images that are recognized by the model as the digit 3 and a cat. In the MNIST experiment, original images with class 3, 5, 7 and 8 are easily crafted into adversarial images recognized as 3. In the CIFAR-10 experiment, the original images with class deer, dog and frog are more easily crafted into adversarial images recognized as a cat. From Figure 7b and Figure 8b, it can be seen that, similar to the targeted attack, the heat map of original–predicted classes of adversarial sample pairs has a symmetrical relationship, especially in the CIFAR-10 dataset.

In MNIST, original images with true label 4 are easier to be crafted into adversarial samples with predicted label 9, and original images with true label 3 are easier to be crafted into adversarial samples with predicted label 5 in untargeted attacks, and vice versa. The reason may be because the digit 4 is similar to the digit 9, and the digit 3 is similar to the digit 5. However, this symmetry is not absolute. Among the 100 original images with class 6, 81 adversarial samples produced in untargeted attacks are predicted to be 5, but only six in the opposite case. Perhaps, in comparison, the handwritten digit 5 and the handwritten digit 3 are more similar than the handwritten digit 6, which to a certain extent shows that the DNN model has indeed learned the characteristics of each class. In the CIFAR-10 dataset, there are cat–dog class pairs that exhibit the same properties, which is consistent with the description in [26].

In untargeted attacks, even the same image can be crafted into adversarial images that are predicted to be a different class to a certain extent. This is due to the randomness of ABCAttack.

### 5.4. Attacking Defenses

While multiple defenses already exist, researchers have been trying to study new attacks that render defenses ineffective. Our proposed ABCAttack can also break some existing defenses. The defense methods we use include adversarial training, input transformation and some overall defense. For all the following attacks against CIFAR-10, we have followed the setting ϵ=8.0 or ϵ=0.031 and MaxEvaluation= 10,000, unless otherwise stated. The experimental results can be seen in Table 3.

Athalye et al. [40] point out that although a “confusion gradient” can defeat some attacks based on gradient optimization, the defense relying on this “confusion gradient” can still be avoided. A BPDA white-box attack is proposed, which can break some defenses based on a “confusion gradient”. We selected some of the mentioned defenses to evaluate the effectiveness of our method for defense strategies.

We first evaluated our attack on defense-GAN [31] using the MNIST dataset and adding $L_{2}$-criteria to the fitness function, and we achieved a success rate of 62.4% and a BPDA of 45%. Stochastic Activation Pruning (SAP) [32] introduces randomness into neural networks to resist adversarial samples. The application of SAP will slightly reduce the accuracy of cleaning classification, but increase the robustness. We evaluated ABCAttack’s performance against SAP defense on the CIFAR-10 dataset. When setting parameters ϵ=0.031, MaxEvaluation= 20,000 and ϵ=0.031, MaxEvaluation= 20,000 and ϵ=0.05, MaxEvaluation= 10,000, the success rate is 64.6%, 67.4% and 88.4%, respectively. Input transformations [33] defend against the adversarial samples by converting inputs. We evaluated the effectiveness of this defense on the ImageNet dataset, and our method can still achieve a 78% success rate in this defense (JPEG and bit depth). Local Intrinsic Dimensionality (LID) [41] is a general metric used to measure the distance from input to adjacent input. The results of the attack on the CIFAR-10 dataset show that our ABCAttack can still reduce the accuracy of the model to 0.01% under the LID defense strategy, and ASR reaches 99.9%.

The focus of [42] is to analyze and evaluate some defenses against adversarial samples in detail, and the study points out that defenses that are claimed to be robust to white-box attacks can still be broken. No attack strategy is sufficient to deal with all defenses, that is, to break a defense, its internal mechanism also needs to be carefully analyzed and adjusted. We evaluated two of these defenses. In *k*-winners defense takes all in [42], we obtained 70.4% ASR in the adversarial training model, and the accuracy of the model was reduced to 0.16% in the adversarial training model in [42]. Pang et al. [43] proposed to train an ensemble of models with an additional regularization term that encourages diversity in defense adversarial samples. We set the maximum $L_{\infty }$ distance $\delta_{max}=0.031$ and $\delta_{\infty }=0.05$ in the CIFAR-10 dataset, and obtained ASR of 71.4% and 96.3%, respectively, when limiting the maximum number of queries to 10,000. When MaxEvaluation= 20,000 and $\delta_{\infty }=0.031$, the ASR is 98.3%.

As can be seen from Table 3, the black-box ABCAttack of this paper can break the above defenses to a great extent, but the attack cost increases. All the above experiments were performed with untargeted attacks. Compared with the attacks of [40,42], the proposed ABCAttack has a lower attack success rate for the defense model, especially the model of adversarial training. In the future, we will further enhance ABCAttack on the defense model and reduce the attack cost.

**Table 3 entropy-24-00412-t003:** Evaluating the effectiveness of ABCAttack on the existing defense strategy.

Defense	Dataset	Parameter Setting	AvgQueries	ASR
defense-GAN [31]	MNIST	$\delta_{max}=0.031$ MaxEvaluation= 20,000	1066	62.40%
SAP [32]	CIFAR-10	$\delta_{max}=0.031$ MaxEvaluation= 10,000	986	64.60%
SAP [32]	CIFAR-10	$\delta_{max}=0.031$ MaxEvaluation= 20,000	1491	67.40%
SAP [32]	CIFAR-10	$\delta_{max}=0.05$ MaxEvaluation= 10,000	701	88.40%
JPEG and bit depth [33]	ImageNet	$\delta_{max}=0.05$ MaxEvaluation= 30,000	4893	78.00%
LID [41]	CIFAR-10	$\delta_{max}=0.031$ MaxEvaluation= 10,000	362	99.90%
*k*-winners [42] (adversarial training model)	CIFAR-10	$\delta_{max}=0.031$ MaxEvaluation= 10,000	937	70.40%
Ensemble train [43]	CIFAR-10	$\delta_{max}=0.031$ MaxEvaluation= 10,000	893	89.50%
Ensemble train [43]	CIFAR-10	$\delta_{max}=0.05$ MaxEvaluation= 10,000	448	96.30%
Ensemble train [43]	CIFAR-10	$\delta_{max}=0.05$ MaxEvaluatio= 20,000	744	98.30%

### 5.5. The Wide Applicability of ABCAttack to Various DNN Models

Following AdversarialPSO, we used HRNN and MLP as target models on the MNIST dataset, which achieved 98.76% and 97.94% accuracy on the test set, respectively, and used CNNCapsule (with data augmentation) and ResNet (ResNet20V1 with data augmentation), which achieved test accuracy of 81.89% and 90.61%, respectively, as target models on the CIFAR-10 dataset to perform untargeted attacks. The networks and training code can be found at: https://github.com/rstudio/keras/tree/master/vignettes/examples (accessed on 4 November 2020). The experimental results are shown in Table 4.

In addition to the target model in Table 4, we also implemented the proposed ABCAttack with different models as target models on MNIST and CIFAR-10 datasets. The experimental results are shown in Table 5. On the MNIST dataset, we performed targeted and untargeted attacks with Lenet5, LSTM and CNN models as victim models. We also used ResNet20V1, ResNet32V1, ResNet44V1, ResNet56V1, ResNet110V1, ResNet20V2, ResNet56V2 and ResNet110V2 as target models on the CIFAR-10 dataset. These ResNet models were trained with data augmentation and, respectively, obtained the following accuracy rates on the CIFAR-10 test set: 90.61%, 91.58%, 91.92%, 92.27%, 91.54%, 91.31%, 92.93%, 92.9%.

It can be seen from Table 4 that under the same target model, our method has a greater advantage in the AvgQueries used to successfully attack. On different target models, the attack success rates of the two methods have their own advantages and disadvantages, but the difference is relatively small. From Table 4 and Table 5, it can be seen that the proposed ABCAttack can obtain superior results on target models with different structures, different datasets and different model sizes. This shows that it has good applicability.

## 6. Discussion

In this paper, the ABC algorithm is used to generate adversarial images. As a swarm intelligence algorithm, the performance of ABC is sensitive to the selection of the hyperparameter of population size. In addition, under the constraint of the maximum change in the image, the value of $\delta_{\infty }$ can also affect the performance of ABCAttack. Therefore, in this section, we will discuss the impact of population size, maximum $L_{\infty }$-norm perturbation and maximum queries on the AvgQueries and ASR.

Population size: A larger population size will increase the diversity of candidate solutions, thereby enhancing the exploration ability of the algorithm. In this paper, a single candidate solution input to the target model is recorded as a query, so we need to set the population size reasonably to balance the algorithm performance and the average queries. As can be seen from Figure 2, we increase the population size from 10 to 30 in steps of 2, and observed the change in the average queries, and found that with the continuous growth of the population size, the average queries used for successful attacks no longer decreased, but remained relatively stable, and even increased slightly. Therefore, we conclude that setting the population size to 20 is a more reasonable choice to balance the convergence speed of the algorithm and the average queries.

$\delta_{max}$: $\delta_{max}$ is the maximum absolute value in a single dimension limited to the added perturbation on the clean image, and it is obvious that $\delta_{max}$ directly affects the adversarial image quality, attack success rate and average queries. As the greater the increased perturbation amplitude, the greater the distance between it and the clean image in the pixel domain, the easier it is to cause the model to give wrong prediction results, and the attack algorithm can obtain higher attack success rate with fewer queries. We show in Table 3 the average queries and success rates of the proposed attack under different $\delta_{max}$ in breaking SAP and ensemble training defense strategies, which prove this. However, correspondingly, the distance between the generated adversarial image and the clean image is larger, and when the value of $\delta_{max}$ increases to a certain extent, the generated adversarial samples may cause humans to fail to distinguish their original class. Therefore, the added adversarial perturbation needs to be constrained to be less different from the original image.

## 7. Conclusions and Future Work

In this paper, a black-box attack using the ABC algorithm ABCAttack is proposed, and related experiments are carried out on MNIST, CIFAR-10 and ImageNet datasets. The results of comparison with methods such as ZOO, C&W, AdversarialPSO, GenAttack and UAP show that the proposed ABCAttack can efficiently generate adversarial images with high attack success rates. We analyze the relationship between SN, α and the $AvgL_{2}$ and AvgQueries of the generated adversarial images. It is noticed that the smaller the $L_{2}$-norm between the initial food sources and the original images, the smaller the final $L_{2}$-norm of the adversarial perturbations. By comparing the original–predicted class of adversarial sample pairs of untargeted and targeted attacks, it is found that some images are easier to be made into such adversarial images, where the predicted label of the adversarial examples and the corresponding true label show some visual or appearance similarity for both a CNN and human eyes. To some extent, this reflects that the DNN models has learned the local information of the class. The proposed ABCAttack can still break some defenses to a certain extent, even if these defenses claim to be robust to white-box attacks. This puts forward new requirements for future defenses, and should be evaluated in black-box attacks. It seems that there is a long way to go for the defense of adversarial samples. The experiments in this paper further illustrate the importance of population-based gradient-free black-box attacks in adversarial attacks, and provide a future research direction for ML system attacks and defenses.

Although the experimental results show the effectiveness of ABCAttack, there is still space for further improvement. The existing GenAttack, AdversarialPSO and ABCAttack are all black-box attacks, they can achieve high attack success rates with fewer average queries and the $L_{2}$-norm of perturbations is larger than that of white-box attacks and gradient evaluation methods. ABC has many variants and improvements in this algorithm. In the next step, we will pursue the use of fewer queries to produce higher-quality adversarial images with a smaller distance from the original images. Secondly, it takes time and effort to craft adversarial samples with high-dimensional data. How to improve the applicability of the algorithm to high-dimensional data needs further research under the premise of maintaining the attack success rate. Finally, the defense ability of many existing defense measures against population-based gradient-free black-box attacks is greatly reduced, which undoubtedly increases the urgency of designing more effective defense measures against attacks. In addition, no single defense can effectively defend against all attacks, and there will always be powerful attacks to break the defense, which makes us more vigilant about the security issues of DNN models. In the future, we will also focus on the root causes and interpretability of adversarial samples, and study more robust defense methods to enhance the security of DNN models.

## Figures and Tables

**Figure 1 entropy-24-00412-f001:**
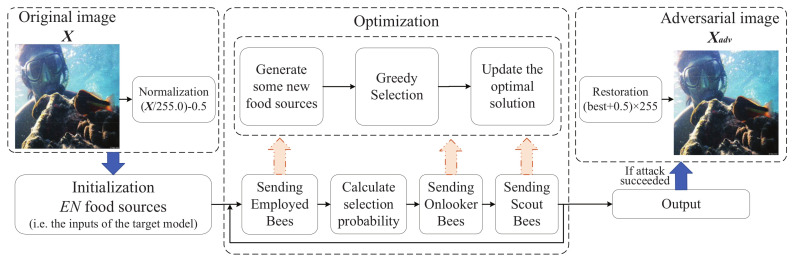
The process of adversarial attacks using ABCAttack.

**Figure 2 entropy-24-00412-f002:**
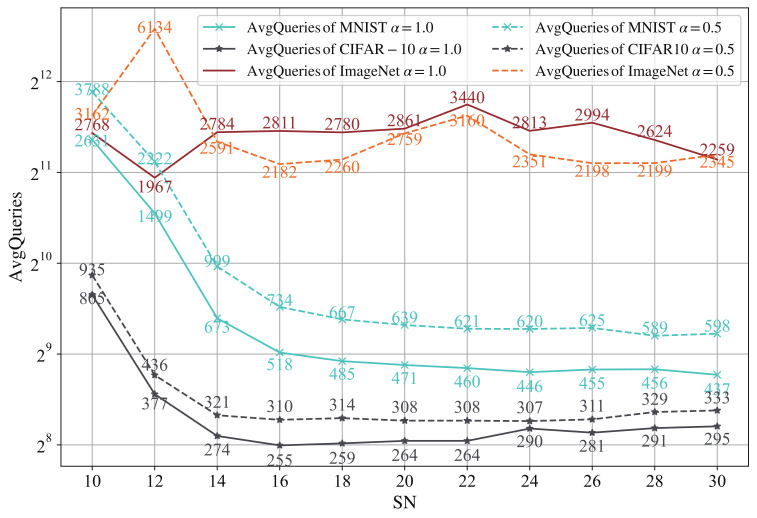
The relationship between the number of food sources and average queries.

**Figure 3 entropy-24-00412-f003:**
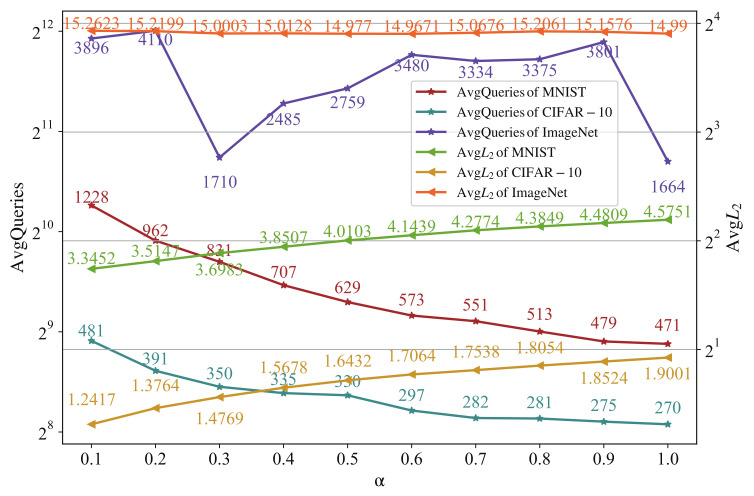
The relationship between α and average queries and $L_{2}$-norm of adversarial perturbation.

**Figure 4 entropy-24-00412-f004:**
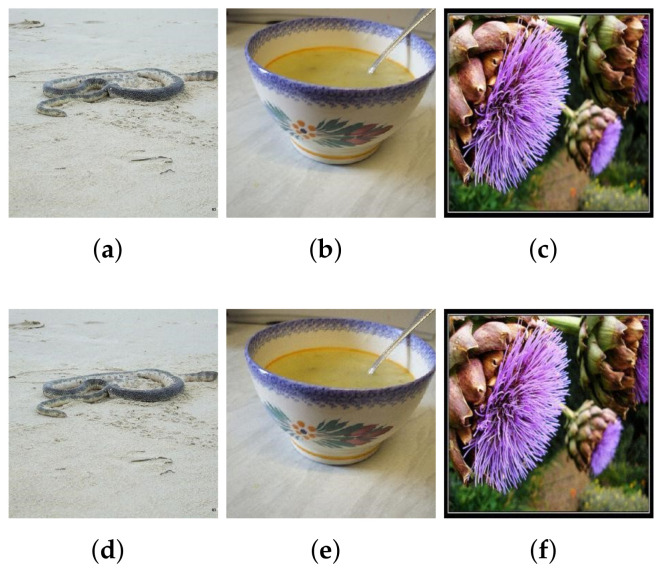
Original samples (**top**) and their corresponding adversarial samples (**bottom**) in ImageNet dataset. (**a**) Predicted label: sea snake. (**b**) Predicted label: soup bowl. (**c**) Predicted label: artichoke. (**d**) Predicted label: Indian cobra. (**e**) Predicted label: red wine. (**f**) Predicted label: cardoon.

**Figure 5 entropy-24-00412-f005:**
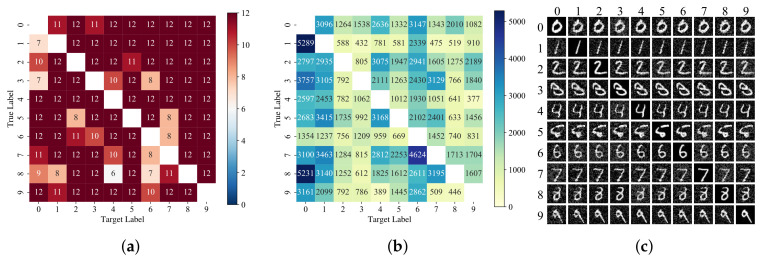
Analysis of targeted attack results in MNIST dataset. The row label is true label and column label is target label. (**a**) Heat map of the number of successful attacks in the MNIST. (**b**) The average queries used by each different original–target class pairs in MNIST. (**c**) The original images in the MNIST test set and the generated adversarial images by ABCAttack.

**Figure 6 entropy-24-00412-f006:**
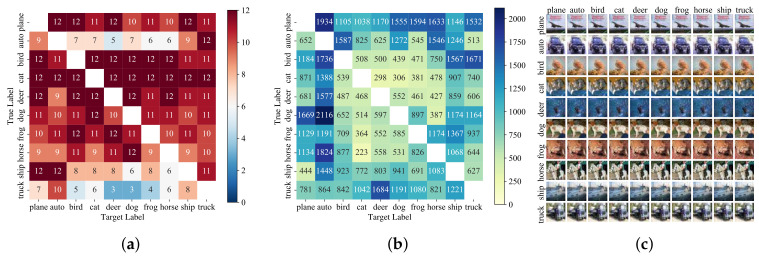
Analysis of targeted attack results in CIFAR-10 dataset. The row label is true label and column label is target label. (**a**) Heat map of the number of successful attacks in the CIFAR-10. (**b**) The average queries used by each different original–target class pairs in CIFAT-10. (**c**) The original images in the CIFAR-10 test set and the generated adversarial images by ABCAttack.

**Figure 7 entropy-24-00412-f007:**
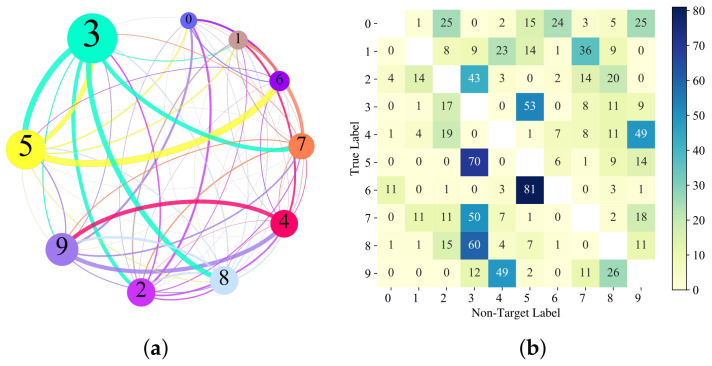
Analysis of untargeted attack results in MNIST dataset. (**a**) The relationship between the label of original images and predicted label of corresponding adversarial images in MNIST dataset. (**b**) The number of successful crafted adversarial images for each class of MNIST dataset in untargeted attacks.

**Figure 8 entropy-24-00412-f008:**
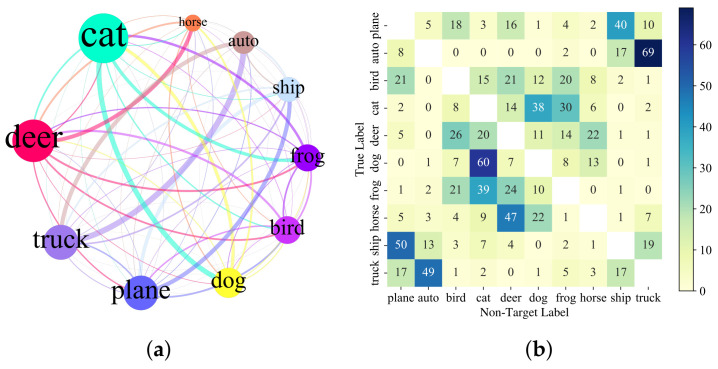
Analysis of untargeted attack results in CIFAR-10 dataset. (**a**) The relationship between the label of original images and predicted label of corresponding adversarial images in CIFAR-10 dataset. (**b**) The number of successful crafted adversarial images for each class of CIFAR-10 dataset in untargeted attacks.

**Table 1 entropy-24-00412-t001:** Result comparison in MNIST and CIFAR-10. The experimental results of ZOO attack are black-box (ZOO-ADAM), and in the experimental results of C&W, the average time is calculated as the total time for training the surrogate model and generating adversarial samples. The bolded item indicates the best result.

MNIST								
	Untargeted				Targeted			
Attack	*ASR*	*Avg* $L_{2}$	*AvgTime*	*AvgQueries*	*ASR*	*Avg* $L_{2}$	*AvgTime*	*AvgQueries*
ZOO	**100%**	**1.4955**	1.38 min	384,000	**98.90%**	**1.987068**	1.62 min	384,000
Black-box (Substitute Model + FGSM)	40.6%	−	0.002 s (+6.16min)	−	7.48%	−	0.002 s (+6.16min)	−
Black-box (Substitute Model + C&W)	33.3%	3.6111	0.76 min (+6.16min)	4650	26.74%	5.272	0.80 min (+6.16min)	4650
GenAttack	−	−	−	−	94.45%	5.1911	−	1801
AdversarialPSO	96.30%	4.1431	0.068 min	**593**	72.57%	4.778	0.238 min	1882
ABCAttack (α=0.5)	**100%**	4.01033	**0.018 min**	629	94.89%	4.7682	**0.048 min**	**1695**
SWISS	**100%**	3.4298	0.087 min	3043	19.41%	3.5916	0.345 min	20,026
ABCAttack (α=0.1)	**100%**	3.34524	0.045 min	1228	94.99%	4.5617	0.051 min	2066
CIFAR-10								
	Untargeted				Targeted			
Attack	*ASR*	*Avg* $L_{2}$	*AvgTime*	*AvgQueries*	*ASR*	*Avg* $L_{2}$	*AvgTime*	*AvgQueries*
ZOO	**100%**	**0.19973**	3.43 min	128,000	**96.80%**	**0.39879**	3.95 min	128,000
Black-box (Substitute Model + FGSM)	76.1%	−	0.005 s (+7.81min)	−	11.48%	−	0.005 s (+7.81min)	−
Black-box (Substitute Model + C&W)	25.3%	2.9708	0.47 min (+7.81min)	4650	5.3%	5.7439	0.49 min (+7.81min)	4650
GenAttack	−	−	−	−	98.09%	1.3651	−	1360
AdversarialPSO	99.60%	1.414	0.139 min	1224	71.97%	2.925	0.6816 min	6512
ABCAttack (α=0.5)	98.60%	1.64319	**0.0233 min**	**330**	82.3%	1.910103	**0.0615 min**	**851**
SWISS	99.80%	2.3248	0.1264 min	2457	31.93%	2.9972	1.623 min	45,308
ABCAttack (α=0.1)	98.40%	1.24167	0.031 min	481	80.88%	1.644	0.0654 min	1102

**Table 2 entropy-24-00412-t002:** Result comparison in ImageNet dataset.

Method	MaxEvaluation = 30,000			MaxEvaluation = 50,000		
	ASR	AvgQueries	AvgTime	ASR	AvgQueries	AvgTime
UAP [29] (VGG-16)	78.3%	-	-	-	-	-
UAP [29] (VGG-19)	77.8%	-	-	-	-	-
UAP-Fast Feature Fool [30] (VGG-16)	47.10%	-	-	-	-	-
UAP-Fast Feature Fool [30] (VGG-19)	43.62%	-	-	-	-	-
Tran-TAP [39] VGG-16 to Inc-v3	76.1%	-	-	-	-	-
Tran-TAP [39] Inc-v3 to VGG-16	70.6%	-	-	-	-	-
Tran-UAP [39] VGG-16-UAP to Inc-v3	68.8%	-	-	-	-	-
Tran-UAP [39] Inc-v3-UAP to VGG-16	60.2%	-	-	-	-	-
ZOO	88.9%	1,280,000	8.031 min	-	-	-
AdversarialPSO	82.00%	2833	3.181 min	-	-	-
SWISS	70.66%	8429	5.014 min	-	-	-
ABCAttack (VGG-16)	99.60%	1839	2.07 min	99.8%	1901	2.1276 min
ABCAttack (VGG-19)	99.00	1501	1.7298 min	99.6%	1698	1.8832 min
ABCAttack (MobileNet-v3)	100%	568	0.6714 min	100%	568	0.6714 min
ABCAttack (Inc-v3, image size is 299)	90.00%	2759	3.172 min	92.00%	2971	3.238 min
ABCAttack (Inc-v3, image size is 224)	98.4%	899	1.0086 min	98.4%	899	1.0086 min

**Table 4 entropy-24-00412-t004:** Comparison of the results of ABCAttack and AdversarialPSO on different target models, and the bolded item indicates the best result.

MNIST				
	HRNN		MLP	
Attack	*ASR*	*AvgQueries*	*ASR*	*AvgQueries*
AdversarialPSO	100%	552	94.70%	548
SWISS	100%	3214	100%	1984
ABCAttack (α=0.5)	100%	**395**	99.40%	**412**
ABCAttack (α=0.1)	100%	1083	99.80%	715
CIFAR-10				
	CNNCapsule		ResNet	
Attack	*ASR*	*AvgQueries*	*ASR*	*AvgQueries*
AdversarialPSO	97.80%	2052	100%	1723
SWISS	98.90%	3725	100%	1792
ABCAttack (α=0.5)	100%	**164**	99.20%	**165**
ABCAttack (α=0.1)	100%	300	99.20%	290

**Table 5 entropy-24-00412-t005:** The wide applicability of ABCAttack on various target models.

Model	Dataset	Untargeted		Targeted	
		ASR	AvgQueries	ASR	AvgQueries
CNN	MNIST	99.50%	381	80.30%	806
LSTM		100%	184	98.50%	920
Lenet5		99.90%	291	93.90%	1648
ResNet20V1	CIFAR-10	99.10%	161	91.60%	798
ResNet32V1		98.80%	140	93.10%	892
ResNet44V1		99.10%	202	91.40%	900
ResNet56V1		99.60%	171	93.20%	617
ResNet110V1		99.50%	180	90.30%	682
ResNet20V2		99.60%	136	95.30%	547
ResNet56V2		99.40%	150	94.30%	546
ResNet110V2		98.90%	141	95.80%	564

## Data Availability

Not applicable.

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
