# Peer review of "ABCAttack: A Gradient-Free Optimization Black-Box Attack for Fooling Deep Image Classifiers"

_entropy, 2022, doi:10.3390/e24030412_

Round 1

Reviewer 1 Report

Regarding this paper, I have some questions that need to be clarified. The paper does a lot of experiments, but I don't understand the important purpose of the paper. Could the authors use more concrete real-world examples to illustrate the purpose of the paper's title?

Artificial Bee Colony (ABC) is a meta-heuristic swarm intelligence optimization algorithm proposed by Professor Karaboga in 2005. This paper proposed ABCAttack using existing datasets: MNIST, CIFAR-10, and ImageNet. The results demonstrated that ABCAttack uses fewer queries and time consuming to generate adversarial samples in Table 2. Please list computational time of ZOO, AdversarialPSO, SWISS, ABCAttack(VGG-16), ABCAttack(VGG-19), and ABCAttack(Inc-v3) under the conditions of the computing machines in your laboratory.

I know that an adversarial example is a modified version of a clean image that is intentionally perturbed (e.g. by adding noise) to confuse a machine learning technique, such as deep neural networks.

  1. Why do we need to generate adversarial samples without the need for a further gradient evaluation and training of the substitute model?
  2. How to establish a set of security tools which can assist users to detect and resist these malicious adversarial example attacks by your proposed methods?
  3. Please also show more detection results (MNIST, CIFAR-10, and ImageNet) after being attacked by adversarial examples under some given deep learning model ()VGG-16, VGG-19, and Inc-v3).

To be confirmed.

The difference between the adversarial samples generated by this attack algorithm and the original samples can’t be perceived by human eyes, as shown in Fig.4. Is it correct that the title of Figure 4. (left and right OR up and down) ?

Author Response

Response to Reviewer 1 Comments

Thank you for your constructive comments and suggestions on the manuscript. Those comments are all valuable and very helpful for revising and improving our paper. After carefully studying the comments the modification of this paper and the response to the reviewers are as following:

Point 1: Regarding this paper, I have some questions that need to be clarified. The paper does a lot of experiments, but I don't understand the important purpose of the paper. Could the authors use more concrete real-world examples to illustrate the purpose of the paper's title?

Response 1:

Let's illustrate from the following example: in the process of extracting numbers from an image (Bank card number recognition and other applications), after the characters in the image are segmented, the digital recognition algorithm (MNIST classifier) is used to identify each character. The attacker can add advetsarial perturbation to the captured image to make the algorithm identify errors. Traffic sign recognition algorithm plays an important role in autonomous driving, but an attacker can use a clean image to generate a corresponding adversarial image. Human beings look almost the same two images, and the traffic sign recognition algorithm will give completely different results, which may lead to safety accidents.

The deep learning algorithm is not robust to the adversarial samples, which not only appears in the image, but also faces the same problem in the field of text and speech. Therefore, the research on the generation of adversarial samples plays an important role in the application of deep learning system, especially the system with strict safety requirements.

This paper studies the effective gradient-free black-box adversarial sample generation method from the perspective of attack. On the one hand, it provides an effective method for the research of adversarial sample generation. On the other hand, it can be used as an evaluation method to test the robustness of deep learning model to adversarial samples, and promote the progress of defense research.

Point 2: Artificial Bee Colony (ABC) is a meta-heuristic swarm intelligence optimization algorithm proposed by Professor Karaboga in 2005. This paper proposed ABCAttack using existing datasets: MNIST, CIFAR-10, and ImageNet. The results demonstrated that ABCAttack uses fewer queries and time consuming to generate adversarial samples in Table 2. Please list computational time of ZOO, AdversarialPSO, SWISS, ABCAttack(VGG-16), ABCAttack(VGG-19), and ABCAttack(Inc-v3) under the conditions of the computing machines in your laboratory.

Response 1:

We modified Table 2 and added the computational time of the algorithm on a single 1080Ti GPU. Our model searches adversarial perturbations in full-pixel space on the ImageNet dataset, so the perturbation added to each clean image is 299×299×3 or 224×224×3 dimensions. Therefore, the attack time cost on the large-scale ImageNet dataset is relatively high compared to the small-scale dataset.

Point 3: I know that an adversarial example is a modified version of a clean image that is intentionally perturbed (e.g. by adding noise) to confuse a machine learning technique, such as deep neural networks.

Why do we need to generate adversarial samples without the need for a further gradient evaluation and training of the substitute model?

Response 3:

Adversarial examples can be generated using substitute model and gradient evaluation, but there are the following problems:

First, the substitute model is trained with the input and output of the target model, so that the decision boundaries of the two are similar, but they cannot be exactly the same. As a result, when the adversarial samples generated on the substitute model attack the target model, the attack success rate also decreases, especially in targeted attacks. Second, training substitute model is feasible on small-scale datasets such as MNIST, but in large-scale datasets such as ImageNet, the cost of training the substitute model is too high, requiring a lot of queries on the target model to train a substitute model. And, the evaluation of gradients on the model still has biases on the true gradients, and the number of queries required is also high when the image size is large, which affects the efficiency of adversarial sample generation. Therefore, training a substitute model or using gradient evaluation on it to generate adversarial examples is usually expensive and low-effective.

While our proposed ABCAttack isn’t require substitute model and is gradient-free, it improves the query and time efficiency.

Point 4: How to establish a set of security tools which can assist users to detect and resist these malicious adversarial example attacks by your proposed methods?

Response 4:

Users can detect and defend against adversarial samples and build security tools by:1) create an adversarial example detector that differentiates between clean and adversarial examples. 2) add the generated adversarial examples to the training set for adversarial training. 3) after the input is changed, the neural network feeds it into the model. The above method can resist adversarial samples to a certain extent, but it cannot completely solve the adversarial samples generated by this method. For example, adding a part of the adversarial samples to the training set for training, the resulting model can still use this method to generate adversarial samples, query costs are likely to rise.

At present, the defense and detection of adversarial samples is also a hot research issue, but there is still no suitable method or security tool that can resist all adversarial samples. How to improve the robustness of the model to adversarial samples is also a problem we are currently studying.

Point 5: Please also show more detection results (MNIST, CIFAR-10, and ImageNet) after being attacked by adversarial examples under some given deep learning model (VGG-16, VGG-19, and Inc-v3).

Response 5:

We have shown more experimental results in the paper to verify the effectiveness of abcattack, and the relevant experimental results have been added to “5.5 The wide applicability of ABCAttack to various DNN models” and “5.3.2. Attacks on Large-scale image dataset ImageNet”. We compare the test results on different models. MNIST model includes different neural networks such as CNN, LSTM and RNN. ResNet models of different sizes are used as the target model to attack on CIFAR-10 data set, and all the experimental results and the calculation of the test results are given. On different types and sizes of target models, the results of our proposed methods are relatively stable.

We also attacked the MobileNet classifier and increased the limit of the maximum number of queries. The relevant experimental results are shown in Table 2.

Point 6: To be confirmed.

The difference between the adversarial samples generated by this attack algorithm and the original samples can’t be perceived by human eyes, as shown in Fig.4. Is it correct that the title of Figure 4. (left and right OR up and down)?

Response 6:

Thank you very much for your careful review of the details in the paper, we have revised the title of Figure 4 and we are sure it should be up and down: Figure 4 Original samples(up) and their corresponding adversarial samples(down) on ImageNet dataset.

In addition to the above modifications, we also read the paper carefully and corrected some sentences and minor errors. In the end, we hope that the revision is acceptable and look forward to hearing from you soon. Once again, thank you very much for your comments and suggestions, they make the revised paper better.

Thank you very much for your attention and consideration!

Reviewer 2 Report

This paper proposes an adversarial attack model using an Artificial bee colony (ABC) algorithm to generate adversarial samples without the need for a further gradient evaluation and training of the substitute model. Overall, the contribution is clear and valid, the paper is written well.

Some statements of the paper need further justification.
1. It states that "ABCAttack can obtain a high attack success rate with fewer queries in the black-box setting".
However, the number of queries is fixed in the experiment (see table 2). It would be good to see more results conducted using various numbers of queries.
Furthermore, it would be good to include some discussions regarding the choice of parameters used in the proposed approach and how they affect the performance, such as population size, the allowed change in the adversarial image (used in the clip function).

2. It states that "ABCAttack can break some existing defenses to a great extent".
This discussion seems to be provided in Section 5.4, it would be good to summarize the experimental results in some tables.

Minor suggestion regarding the paper writing.
1. Figure 1: It would be good to provide the full name (e.g., EB, OB, and SB) in the figure for the convenience of reading.
2. Do you think it would be good to include ABC in the paper title?

Author Response

Response to Reviewer 2 Comments

Thank you for your constructive comments and suggestions on the manuscript. Those comments are all valuable and very helpful for revising and improving our paper. After carefully studying the comments the modification of this paper and the response to the reviewers are as following:

Some statements of the paper need further justification.

Point 1: It states that "ABCAttack can obtain a high attack success rate with fewer queries in the black-box setting". However, the number of queries is fixed in the experiment (see table 2). It would be good to see more results conducted using various numbers of queries.

Response 1:

We conducted experiments with a greater query limit MaxEvaluation=50,000 on VGG-16, VGG-19, Mobilenet-V2 and Inception-v3. The experimental results have been shown in the Tab.2.

From the perspective of attack success rate, even if the termination conditions of ABCAttack, the maximum query MaxEvaluation has increased by 20000, the increase in attack success rate is also only slight. when the maximum query is increased to 70,000, very few samples still cannot obtain corresponding adversarial samples. Compared with the maximum limit change δmax, increasing the maximum limit query MaxEvaluation will not significantly improve the attack success rate, and will lead to an increase in the average query.

Similarly, we intentionally increased the number of queries when attacking the defense strategy, and the results can be seen in Table 3, which is consistent with the results on ImageNet.

Point 2: Furthermore, it would be good to include some discussions regarding the choice of parameters used in the proposed approach and how they affect the performance, such as population size, the allowed change in the adversarial image (used in the clip function).

Response 2:

We have added “Section 6 Discussion” to the paper to discuss the impact of two hyperparameters, population size and the maximum change allowed in adversarial perturbation .

  1. Discussion

In this paper, the ABC is used to generate adversarial images. As a swarm intelligence algorithm, the performance of ABC is sensitive to the selection of the hyperparameter of population size. In addition, under the constraint of the maximum change of the image, the value of δmax can also affect the performance of ABCattack. Therefore, in this section, we will discuss the impact of population size, maximum -norm perturbation and maximum queries on the AvgQueries and ASR.

Population Size: A larger population size will increase the diversity of candidate solutions, thereby enhancing the exploration ability of the algorithm. In this paper, a candidate solution singe input to target model is recorded as a query, so we need to set the population size reasonably to balance the algorithm performance and the average queries. As can be seen from Fig.2, we increase the population size from 10 to 30 in steps of 2, and observed the change in the average queries, and found that with the continuous growth of the population size, the average queries used for successful attacks no longer decreased, but remained relatively stable, and even increased slightly. Therefore, we conclude that setting the population size to 20 is a more reasonable choice to balance the convergence speed of the algorithm and the average queries.

δmax: δmax is the maximum absolute value in a single dimension limited to the added perturbation on the clean image, it is obvious that the δmax directly affects the adversarial image quality, attack success rate and average queries. Because the greater the increased perturbation amplitude, the greater the distance between it and the clean image in the pixel domain, the easier it is to cause the model to give wrong prediction results, and attack algorithm can obtain higher attack success rate with lower queries. We show in Tab.4 the average queries and success rates of proposed attack under different δmax in breaking SAP and ensemble training defense strategies, which proves this. However, correspondingly, the distance between the generated adversarial image and the clean image is larger, and when the value of δmax increases to a certain extent, the generated adversarial samples may cause humans to fail to distinguish their original class. Therefore, the added adversarial perturbation needs to be constrained to be less different from the original image.

Point 3. It states that "ABCAttack can break some existing defenses to a great extent".

This discussion seems to be provided in Section 5.4, it would be good to summarize the experimental results in some tables.

Response 3:

Following your valuable comments, we have given the experimental results of some existing defenses in table, and we have added more detailed experimental results, which makes it easier for readers. The tables we added in the paper are as follows, and individual sentences have been modified to make the paper easier to read:

Defense

Dataset

Parameter setting

AvgQueries

ASR

defense-Gan

MNIST

δmax =0.3

MaxEvaluation=20000

1066

62.4%

SAP

CIFAR-10

δmax =0.031

MaxEvaluation=10000

986

64.6%

SAP

CIFAR-10

δmax =0.031

MaxEvaluation=20000

1491

67.4%

SAP

CIFAR-10

δmax =0.05

MaxEvaluation=10000

701

88.4%

JPEG & Bit depth

ImageNet

δmax =0.05

MaxEvaluation=30000

4893

78%

LID

CIFAR-10

δmax =0.031

MaxEvaluatio = 10000

362

99.9%

k-winners(adversarial training model)

CIFAR-10

δmax =0.031

MaxEvaluation = 10000

937

70.4%

ensemble train

CIFAR-10

δmax =0.031

MaxEvaluation = 10000

893

89.5%

ensemble train

CIFAR-10

δmax =0.05

MaxEvaluation = 10000

448

96.3%

ensemble train

CIFAR-10

δmax=0.05

MaxEvaluation = 20000

744

98.3%

Minor suggestion regarding the paper writing.

Point 4. Figure 1: It would be good to provide the full name (e.g., EB, OB, and SB) in the figure for the convenience of reading.

Response 4:

Thank you very much for your suggestions. We have modified the Figure 1. For EB, OB and SB, we have adopted the full name for explanation, which makes it more convenient for readers to read Figure 1.

As images can no longer be inserted here, we would like you to view Figure 1 in the revised paper or in the response PDF file.

Point 5. Do you think it would be good to include ABC in the paper title?

Response 5:

Your requirements for the title of the paper are very reasonable. We have reconsidered the title of the paper carefully and included ABC in the paper, which will give readers a general understanding of the work done in this paper when they see the title. We finally changed the title to: ABCAttack: A Gradient-free Optimization Black-box Attack for Fooling Deep Image Classifiers.

In addition to the above modifications, we also read the paper carefully and corrected some sentences and minor errors. In the end, we hope that the revision is acceptable and look forward to hearing from you soon. Once again, thank you very much for your comments and suggestions, they make the revised paper better.

Thank you very much for your attention and consideration!

Round 2

Reviewer 1 Report

The authors have answered all my questions sincerely, and I suggest that this modified version is acceptable to be published.